# “Moving for My Baby!” Motivators and Perceived Barriers to Facilitate Readiness for Physical Activity during Pregnancy among Obese and Overweight Women of Urban Areas in Northern Taiwan

**DOI:** 10.3390/ijerph18105275

**Published:** 2021-05-15

**Authors:** Yvonne Hsiung, Ching-Fang Lee, Li-Kang Chi, Jian-Pei Huang

**Affiliations:** 1Department of Nursing, Mackay Medical College, New Taipei City 25245, Taiwan; yvonnebear@mmc.edu.tw; 2Department of Physical Education and Sport Sciences, National Taiwan Normal University, Taipei City 10610, Taiwan; lchi@ntnu.edu.tw; 3Department of Obstetrics & Gynecology, MacKay Memorial Hospital, Taipei City 104217, Taiwan; huangjianpei@yahoo.com.tw

**Keywords:** physical activity, obese, overweight, pregnant women, focus group

## Abstract

Low levels of physical activity (PA) are of a health concern among high body mass index (BMI) women living a sedentary lifestyle and being overweight or obese during pregnancy is associated with increased risks of maternal and fetal health complications. Obstetricians often provide advice regarding recommended PA levels, yet this has not been easily achieved in this group to prevent adverse birth-related outcomes. The purpose of this study is to explore motivators/enablers and perceived barriers through in-depth qualitative inquiry, guided by a behavioral change model, for understanding of pregnant women’s decisions to engage, or refrain from PA practice. Thirteen overweight and obese pregnant women aged 28 to 45 years with an inactive, sedentary lifestyle in urban areas of northern Taiwan were recruited to participate in six focus group sessions for their intent and readiness for PA engagement in pregnancy. A thematic content analysis was performed with a constant comparison method to categorize interview data and generate themes. The findings illustrate the extent to which obese and overweight pregnant women’s readiness for PA is affected by multiple factors, including personal beliefs, perceived societal norms, peer support, and the competing priorities in the environment. PA interventions are to be effective by focusing on overcoming barriers, increasing motivations, and enhancing self-management. Strategies shared by participants shed lights for program developers to design preferable behavioral interventions for this group of women who are low self-esteem with low self-efficacy to increase PA and meet recommended levels. There is considerable potential for health care providers to provide accessible information, facilitate PA, and promote an active lifestyle during and after pregnancy.

## 1. Introduction

While obesity is found as a significant contributor to chronic diseases [1], women with their body mass index (BMI) above 25 kg/m^2^ [2] at their first antenatal visit are likely to have more clinical implications than their counterparts with normal BMIs during pregnancy and childbirth [3,4]. Over the last two decades, the number of overweight and obese pregnant women have doubled [5,6], allegedly reaching from 40% to 60% in the United Kingdom, Australia, and the United States [7]. According to the Report of Nutrition and Health Survey in 2020 [7], more than 81% of the adult women in Taiwan have their waist circumference, a predictor of metabolic syndrome, larger than 30 cm; particularly Taiwanese women residing in urban areas are either overweight (31.5%) or obese (25.5%). This prevalence of maternal obesity escalates to pose a serious public health concern in Taiwan.

A great amount of maternal obesity research shows that poor management of the gestational weight has resulted in multiple adverse maternal and neonatal outcomes, including unfavorable consequences [3], such as increased chances of miscarriage, gestational diabetes mellitus (GDM), preeclampsia, and preterm birth [8,9], and relevant infant risks, such as perinatal death, congenital anomalies, macrosomia, and other birth injuries [10,11]. In light of the fact that reversing maternal obesity prevents most of its harmful effects for immediate maternal health concerns, interventions have recently emerged to effectively address the epidemic issues of maternal obesity [4] as a global priority [2,12].

For women of all weight ranges, physical activity (PA) has been identified in related research as a key modifiable lifestyle factor [2] to enhance birth-related outcomes. While pregnant women seem to become more receptive to behavioral change interventions [2], There has been increasing interests to develop interventions for an active lifestyle during pregnancy, and among them, inactive overweight and obese women’s leisure-time PA and their daily energy expenditure have become a research emphasis [13]. The 2015 guidelines published by the American College of Obstetrics and Gynecology (ACOG) [14] suggest that pregnant women should continue PA, particularly those who have no medical complications are preferably to engage in moderate-intensity aerobic activity for at least 30 min daily or 150 min per week [2,7,15,16,17]. Based on such scientific evidence that moving more and sitting less have enormous benefits, Taiwanese pregnant women are advised at a national level to maintain recommended PA throughout pregnancy [18,19].

Although there exists an urgent need for public health efforts to facilitate PA in women with high BMIs, the effectiveness of PA interventions is conflicting [2]. PA may be perceived as an unsafe behavior [20] despite the many beneficial mother and infant outcomes, particularly in the early and late pregnancy [21]. Such reluctance to increase PA is wildly documented in the literature that in a recent review found only a small percentage (less than 16%) of American women choose to engage their leisure time in PA during pregnancy [22]; similar findings were revealed in an Irish cohort study that only 6.4% overweight and obese pregnant women have met the national recommendations for exercise—whereas women with a healthy BMI have twice their adherence to PA guidelines [23]. Sui and his colleagues in their thorough literature review argue that not all PA guidelines for pregnant women are based on high-quality evidence [7], hence, the suitability of international PA guidelines [17,24] for Taiwanese women during pregnancy remains unclear. Behavioral change interventions successful for pregnant women with health BMIs may not be as beneficial if applied to the overweight and obese [2,20], and the recommended PA levels may not address specific challenges overweight or obese women face in everyday routines.

Up to now, there has been a dearth of research exploring obese and overweight women’s perceptions of PA during and after pregnancy in Taiwan, and little is known about culturally appropriate strategies and advice preferred from obstetrician care professionals to promote PA. A focus group study attempts to fill in this knowledge gap from perspectives of obese and overweight pregnant women in northern Taiwan. Specific aims are to provide qualitative insights into motivators, environmental enablers, and perceived barriers of PA. In addition, how PA advice from obstetricians may be positively perceived by sedentary overweight and obese participants are to shed lights for program developers to design effective interventions not only during pregnancy but also postpartum.

## 2. Materials and Methods

### 2.1. Design and Participants

This focus group qualitative study was conducted from July to October of 2019 after the research protocol and consent forms were approved by the Human Research Protection Institutional Review Board (# 17MMHIS171). The clinically diagnosed obese and overweight participants included those: (1) who were aged over 20 years, (2) who were able to speak and write Mandarin Chinese, (3) who had their BMI ≥ 25 kg/m^2^ at antenatal booking and for at least 24 gestational weeks, and (4) who had never participated in any PA intervention program before this study. Excluded were pregnant women at high risks of obstetric or medical complications, such as insulin-dependent or gestational diabetes mellitus, hypertensive conditions, and heart-related diseases. During the data collection period, pregnant women from two prenatal outpatient clinics in urban areas of northern Taiwan who met the recruitment criteria were purposively selected by a research coordinator. A total of 15 pregnant women were approached, however, one not only had regular PA during pregnancy but her activity intensity was nearly at the recommended level, and the other women rejected to participate this study due to unavailability and time conflicts. These two were excluded from the study, leaving thirteen qualified subjects in our final sample (a decline rate: 2/15 = 13.3%). It is worth mentioning that all participants in the focus group sessions had some but irregular PA experience during and before pregnancy. Their insufficient PA was generally less than 20 min and fewer than three times per week, making these women appropriate candidates to share perceptions of main barriers and facilitators for future PA engagement. Information about the study goals and procedures was explained in each focus group session; written informed consents were obtained onsite from all the participants with certain sensitivity to protect subjects’ confidentiality.

### 2.2. Focus Group Interviews

A total of three focus groups of six sessions were conducted among qualified participants; interviews were held in a consultation room nearby the obstetrician clinic of two participating teaching hospitals. The three focus groups were composed of five, five, and three obese and overweight women (N = 13), respectively. Each participant was asked to participate in two consecutive sessions, each lasting 40 to 60 min with a short break in between. All sessions were facilitated by the principal investigator of this study with a co-moderator who served as an observer, field note taker, and session assistant. Both moderators were nurse researchers who have pregnant experience and moderated several focus groups of similar topics in collaboration with the participating clinics in an urban community setting.

Semi-structured interview questions were formerly developed (Table 1) based on relevant PA references and core concepts of the Transtheoretical Model (TTM) [25,26], particularly with an emphasis on barriers to and readiness for PA. Following greetings, procedures related to study consents, and explanation of interview ground rules, participants with maternal obesity were invited during the first sessions to freely express their feelings regarding their health status, pregnancy-related concerns, weight management in general, and their PA experiences before and during pregnancy. Most importantly, participants were asked to describe their intent to engage in higher levels of PA and the likelihood to maintain PA postpartum. During the interviews, topics were discussed in the order in which they arose naturally in the sessions [27]. Since a majority of participants shared their experience of seeking pregnancy-related information, in the subsequent session, we placed an emphasis on exploring possible strategies in PA interventions, particularly their perceptions of PA being supported by technology and/or preference of using mobile-Health APPs for PA. All focus group discussions were audio-recorded and subsequently transcribed verbatim to accurately represent the participants’ perspectives, and detailed notes were taken regarding significant responses and important incidents during the group discussions.

### 2.3. Data Analysis

According to the Consolidated Criteria for Reporting Qualitative Research [28], to explore the rich qualitative data collected from interview sessions, the complete set of six group transcripts was independently reviewed, text-coded, and compared by two trained nurse researchers serving as coders, including the principal investigator/primary moderator. A thematic content analysis was employed using inductive comparative methods by both coders [29], with the TTM concepts of readiness for PA in mind. First, each transcript was initially examined for clarifications of unclear texts and an overall picture of the focus group flow. Secondly, individual themes were assigned by both coders with texts/examples to describe the most prevalent response they perceived, such as repeated phrases and emerged patterns in the sessions. As part of an iterative coding process [29], after four focus group sessions were completed, the two coders categorized and aggregated themes into higher levels of concepts, and coders later met to discuss and compare rationales for their assignments. A total of twelve out of fourteen themes were identical, reaching an initial agreement of 86%. For the codes/themes that were not firstly and collectively identified (2/14, 14%), consensus was later reached after deliberations and further examination of the detailed texts/quotes related to these two themes.

At this phase, a comprehensive list of identified themes was compiled for both coders to review the recurring themes and to confirm data saturation. While no new codes emerged in the final two focus group sessions, fourteen themes representative of five concepts at a higher level were finalized by both coders in full agreement. In addition, an independent investigator with qualitative research expertise examined random texts from four samples (about 25% of the interview transcripts) were examined for accuracy and code assignment. Quotes from the transcripts are under alias names in our final result without participants’ identifications in order to protect participants’ confidentiality.

## 3. Results

### 3.1. Demographics

The thirteen overweight and obese pregnant participants (mean age = 30.53 ± 4.57) were predominantly married, nulliparous, and unemployed with college diplomas; residing in metropolitan and urban areas, they self-reported to be fairly sedentary and familiar with APPs on smartphones (Table 2). All obese and overweight women have been suggested by their primary obstetrician to carefully manage gestational weight gain, particularly, to abide the national recommended PA levels [18].

### 3.2. Perceptions of and Readiness for Being Active

All emerging themes are listed in Table 3 and Figure 1 details the concept of overweight and obese women’s readiness for PA during pregnancy. Almost all participants regarded PA a task that requires additional efforts; it was perceived as actions or behaviors that would “*make you sweat*” [Angie 20 weeks, pre-BMI = 33.6 kg/m^2^]. One Chinese (Mandarin) word stood for all three ideas of PA, exercise, and fitness, but the term PA was referred to something “*light and doable”* (Angie) from participants’ perceptions. Exercise and fitness were viewed as much higher levels of PA, such as swimming, biking, fast-walking, and jogging. “*I am pretty sure when my doctor asks me to increase PA, she means exercise, not just take out the trash or do the laundry*” [Sarah, 28 weeks, pre-BMI = 33.3 kg/m^2^]. However, the line between PA and exercise was somewhat obscure.

All participants were familiar with the common advice from obstetricians to increase PA for weight management in pregnancy. However, this advice has been viewed as a cliché (Theme 1)—this was something they have heard for multiple times, since no participants have successfully managed their weight by PA alone, even before pregnancy. This advice from the health care professionals just triggered previous negative experience related to their unsuccessful weight management (Theme 2). “*I have been fat my whole life. Of course, I know I need to exercise in order to lose (weight). They think I don’t know?*” [Amy, 15 weeks, pre-BMI = 27.7 kg/m^2^].

In addition, this PA advice was not perceived any easy by obese and overweight women, since it quickly linked to a mental image that PA meant “*working really hard*” (Amy) or “*making efforts to go to the fitness room*” (Sarah). Negative emotions were provoked with immediate thoughts of multiple obstacles. For example, they felt they were again given with extra burdens and an impossible task to control weight even when carrying a baby (Theme 3). Such expectations from the obstetrician authorities have made pregnant women very vulnerable. “*Lose weight and exercise… blah, blah, blah. I know I sound like I have 100 excuses, but give me a break? I am now pregnant and I feel so tired and so weak…people gave me so many expectations”* [Mary, 35 + 3 weeks, pre-BMI = 35.1 kg/m^2^].

In general, our obese and overweight participants exhibited a low level of readiness for PA, despite the health benefits of PA in pregnancy were well-recognized, especially for women with obesity (Theme 4). However, this personal belief or motivator did not transform into a stronger intention for behavioral change. Also since engaging in PA guaranteed no immediate weight result, overweight participants chose to take obstetricians’ PA advice lightly. The majority of the pregnant women (*n* = 8) admitted that it was unlikely for them to drastically change their current sedentary lifestyle—they were not prepared to become more active any sooner or engaged in PA within six months.

Expressions of low self-esteem were prevalent among the PA contemplators; their situational confidence (low self-efficacy) to behavioral changes were also low. “*I guess I have been low (in self-esteem) due to my (fat) look; being big is an unremovable label since I was very young. I don’t believe one day I would become a completely different person*” [Eve, 35 + 2 week, pre-BMI = 32.4 kg/m^2^]. “*I do NOT have faith in myself. I never succeed. I don’t think exercise is my thing. I just have to accept the fact that I will be like this my whole life*” [Van, 24 + 1 weeks, pre-BMI = 27.6 kg/m^2^].

### 3.3. Motivations to Move for Unborn Babies

It was worth noting that despite participants’ general intention to engage in PA was weak for weight loss, almost all expectant mothers showed a higher level of PA motivation for their unborn babies’ health (Theme 4). This intention to transit from a sedentary lifestyle was associated with better fetus growth, enhanced birth outcomes, and improved maternal health, particularly in hope with a healthier and longer life to fulfill child-bearing responsibilities. Most importantly, expectations from their loved ones, such as significant others of spouses and/or parents, also motivated them to become physically active (Theme 4). “*Of course, the most important thing is to smoothly deliver my baby; I hope I stay healthy enough to play with him for a long time*” [Mary, 35 + 3 weeks, pre-pregnancy BMI (pre-BMI) = 35.1 kg/m^2^]. “*Not only I worry (about a rough delivery), but also my parents worry so much about my weight during the entire pregnancy. They know I haven’t been able to control (my weight) well, and diseases like hypertension and thyrodism run in my dad’s side of family. I need (to increase) PA to get this weight off their mind”* [Eve, 35 + 2 weeks, pre-BMI = 33.6 kg/m^2^]. “*I have fatty liver already. I may not be here long enough…I exercise so that I may see my daughter grow up*” [Babs, 33 weeks, pre-BMI = 26.1 kg/m^2^].

All participants (*n* = 13) were familiar with this saying, “a tongue never gets things done,” and their preparation for PA included: searching online for fitness class information, buying yoga clothes or swimming suits, arranging time for daily exercise, obtaining aerobic activity videos at home, etc. A slight to moderate embarrassment was shown during the interviews that while participants have made their minds to engage in PA, their “inner-oaths” or “advanced announcements to others” never accomplished. Many (4/13, 30.7%) stayed prepared for a fairly long time, and some (*n* = 4) took decades in contemplation, “*I lie on the bed most of my (pregnancy) time, although I know I should get up and start walking a little bit*” (Jane).

Approximately 38.5% of the participants (*n* = 5) have increased their PA after pregnancy, neither strenuous exercise nor intense fitness/sport were preferred (Theme 5). Many chose to adopt a new lifestyle, such as walking more, climbing stairs more often, and starting a gentle workout routine of weight-lifting or slow jogging. “*I started going to the community gym (after pregnancy). I do not run, just walk faster than my usual pace…to increase my heart rate and sweat a little bit…Daily 30 min is enough”* [Jane, 29 + 4 weeks, pre-BMI = 29.3 kg/m^2^]. “*I walk home now. This constitutes my daily exercise*” (Mary). “*I try not to use the escalators. I force myself to walk up the stairs but I have to rest every 2–3 floors”* (Van).

Women who have successfully engaged in leisure-time PA also have intentionally increased the frequency to do housework, visit relatives and friends, window-shop, dine-out, or even work and babysit (other kids) for a longer time. “*I would walk to my sister’s. I ask my husband to take me out for a walk every night after dinner. Go out more for shopping. I want to keep myself moving, you know?”* (Babs). “*I do those pregnancy-appropriate things more often now, like stretching, walking, and deep-breathing*” (Angie).

According to the TTM definition, none of the pregnant women were in the maintenance stage, since none have successfully changed their sedentary lifestyle for longer than six months. The willingness to maintaining a new PA regimen was pervasive for the subgroup in action stage (*n* = 5) (Theme 6). According to these women with higher readiness, their motivation to continue PA seemed to be highly associated with successful experience (confidence in present or past ability) in gestational weight management. Unlike their counterpart with less readiness, the perceived and experienced PA benefits were not only for their unborn baby but also for themselves. “*I see this (exercise) works and I want to continue doing it. I just hope I have enough time*” (Jane).

### 3.4. Multiple Barriers to Engage in PA

Barriers to PA engagement shared by participants could be categorized into personal, social, and environmental dimensions. The most prevalent theme of barriers was a personal belief in the need to limit their PA in order to prevent miscarriages (Theme 7). All Taiwanese participants were familiar with the traditional saying, “In your pregnancy, sitting rather than standing, and lying down rather than sitting.” Many agreed they were “not supposed to move,” particularly in early pregnancy. Limiting PA in pregnancy was a direct request (or viewed as an order) from senior family members; in Taiwanese culture, mothers-in-law’s opinions to limit PA had more weight on women. Participants also believed that PA might trigger severe uterine contractions, aggravate existing muscle-skeleton injuries, or induce other unexpected medical condictiones they could not control (Theme 8). The thought to possibly hurt themselves or lose their unborn babies have jointly hindered readiness for PA in their decisional balance, despite the well-recognized benefits of PA. “*I get heavy breaths (from PA); I might pass out in nowhere”* [Bella, 38 weeks, pre-BMI = 25.1 kg/m^2^].

Other salient barriers to engage in PA during focus group sessions were related to the unsupportive environment, including: low support from peers and significant others, external conditions inappropriate for PA, and a long list of competing family and work priorities (Theme 9). Common external barriers perceived to decrease PA motivations included: “*weather being too hot to work out*”, “*PA drained energy for the day*”, “*Got beat up already from work and/or commute*”, “*duties and burdens from babysitting (other children) or taking care of older parents*”, “*no peer members to encourage each other or exercise together*”, “*unsupportive husbands or family members*”, “*schedule conflicts*”, “*no me-time allowed for exercise*”, etc. An obese participant stated: “*My sister and I talked about going to swim together, but she had a car accident. My husband would never exercise with me. I hate to get sticky (in summer). Lots of meetings at work. I come home late and I only have 10% energy left to get on the bus…just many reasons. I know I sound lazy*” (Mary).

### 3.5. Preference of PA Advice from Obstetricians and Strategies for the Overweight and Obese

Our moderator(s) also solicited suggestions about how participants preferred to receive PA advice from obstetrician professionals; most importantly, how these overweight and obese women could be motivated or assisted to become physically active. A pervasive theme related to PA advice was participants’ preferred communication style from obstetricians. Although participants admitted receiving a great amount of information during prenatal visits, a need was expressed to have a more empathetic, two-way communication with their obstetricians. Conversations with the obstetrician professionals were generally described fairly unidirectional. While PA and exercise was frequently brought up for gestational weight management, women with obesity felt being passively “weight-watched” throughout the entire pregnancy course. “*When every time they said to lose weight or exercise more, I felt like a child……it’s not that I don’t want to listen to them. They were not us. I deserve some empathy”* (Babs). The professional advice to increase PA should be given with personal touches (Theme 10).

Since all participants’ highest priority was to safely and smoothly deliver their babies, a “fetus-centered” approach was a preferred strategy (Theme 11) that advice should be based on fetus’ growth and development. “*I would follow (doctor’s advice) better if it’s for my baby… I guess this is my mother nature*” (Van). On the other hand, to better facilitate readiness among pregnant women with high BMIs, individually tailored assistance was required. A one-on-one PA consultation conducted by health care experts would assess individual needs and eliminate multiple personal and environment barriers. “*We all have different difficulties and one plan doesn’t fit all. A tailor-made PA program would better help me come up with an achievable, daily exercise routine*” (Bella).

### 3.6. Peer Support during Focus Group

An unexpected theme emerged and participants provided support for each other during all interview sessions. Unrelated to the study purpose, our focus group served as a source of support from this peer group with low self-esteem and low self-efficacy. While weight management topics were brought up, participants who were complete strangers met for the first time could quickly relate to these emotions and instantly offered empathy and support with no preservation. Their comments ranged from simple acknowledgements such as, “*I am sorry to hear that*”, “*Oh, I know, I feel for you*” and “*pat-pat*” to lengthy experience sharing (of being judged as “lazy and ugly”) and/or a personal story of life-time struggles against weight gain. Collectively, these expectant mothers shared their joys and concerns related to pregnancy, yet many also expressed frustrations and negative experience of long-time weight management, such as their mind battles with weight gains (Themes 12–14). Self-depreciation and a sense of humor were observed as a reflection of these obese and overweight women’s body image. “*If I faint, nobody could carry me*” (Bella).

## 4. Discussion

### 4.1. A Theoretical Basis of Perceptions and Readiness Stages for PA

This focus group study is framed based on the Transtheoretical Model (TTM) [25,26], originated from the Theory of Planned Behavior (TPB) [30] and other health behavioral theories. Our qualitative data collection, analysis, and results are structured and reported according to the core concepts of the TTM in which participants’ readiness predicts their PA behavioral change within six months. Pregnant women’s perceptions, personal beliefs, external cultural and normative beliefs are to possibly mediate their intentions to change. Although the general suitability of the TTM theoretical basis requires further examination among Taiwanese pregnant women, the utility of core TTM concepts in the planning of PA interventions among mothers has been widely confirmed [31].

Using a theoretical approach has allowed us to examine PA readiness through beliefs, attitudes toward, and intentions to behavioral change. In previous PA research, various theories have been utilized to explain and predict PA during pregnancy, however, most studies were criticized for only focusing on the intrapersonal level determinants, including the commonly used TPB [30]. In our study, the important role of social support has not been omitted within the context of Taiwanese obese and overweight pregnant women’s perceptions and decision-making. We are confident that the TTM and our concept map appropriately address both intrapersonal and social factors to facilitate pregnant women’s readiness for PA.

While our fourteen themes presented in Figure 1 has mapped directly on to the TTM concepts, it is worth noting that we have no intention to create a new theoretical model to explain our theoretical perspective for conducting this study. We merely attempt to depict all emerging themes (listed in Table 3) in one figure so that overweight and obese women’s readiness for PA and obstetrician care providers’ stage-wise facilitation during pregnancy could be easily comprehended by our readers. In particular, this graphical presentation also details pregnant women’s motivations that they could be strategically raised by preferred PA advice in effective interventions and supported by peers when multiple individual and family related barriers are adequately addressed and overcome.

### 4.2. Readiness for PA Engagement

At the beginning of the focus groups, patients were inquired to describe the mental link between their weight and PA, and what “PA, exercise, and fitness” meant to them. Our rationale is to gradually introduce these topics for an overall understanding of images or thoughts evoked by these words frequently used in obstetricians’ PA advice. The emerged themes are comparable to findings in an Australian cohort study [31] and a review article investigating obese pregnant women’s perceptions of being active during pregnancy [7]. For women of high BMIs, their negative emotional arousal is expected to adversely influences pregnant women’s self-efficacy. Not only a sedentary lifestyle affects psychosocial resources through negative emotions [32] but also the unsuccessful weight management have linked to previous frustrations, makes contemplating PA difficult in this women group.

Even though the immediate emotional arousals were mostly undesirable, as believers of PA benefits [33], our participant recognized the importance of PA in pregnancy, such as improved cardiovascular condition and glucose tolerance, bone and muscle mass, and reduced risks of obesity and birth complications [7]. To be more specific, participants know they are the high-risk group for adverse obstetrical outcomes and need PA more than their counterpart with healthy BMIs. PA advice during clinical visits is considered an important part of prenatal care, and participants provided examples of modest PA, perceived appropriate in pregnancy.

According to the TTM, individuals with great intentions and external support have better potential to change. Our findings also confirm that participants’ readiness for PA is stage-wise: those have successfully increased their PA (*n* = 5) also have transited from a preparation to action stage; this subgroup also exhibits a better sense of control in managing gestational weight. This finding is consistent with results from a recent randomized controlled trial [34] that through PA facilitation, obese women reduce a mean of 1.38 kg during pregnancy (*p* = 0.040). Barriers and enablers (motivations to increase intentions) are important factors in high BMI women’s decisional balance, but up to now, readiness for PA and its correlates are still unclear in Taiwan [35]. Our study, excluding women with clinical diagnoses of depression and/or other psychiatric disorders, reveals that less than 40% of this sedentary group of high BMIs engage in light to moderate PA and none of them meet PA guidelines of 150 min or more.

The majority admit they are unclear about the national recommended PA guidelines. This level of knowledge is comparable to what are reported in a correlational study [35] among obese and overweight pregnant women whose BMI (34.13 ± 7.07 kg/m^2^) and weeks of gestation (15.68 ± 2.44). We have the confident to conclude that rather than personal belief and attitudes toward PA benefits, obese and overweight women’s readiness to meet the PA guidelines are affect by their awareness and self-efficacy [7]. It may be more beneficial to this group if PA’s role in increasing energy and thus enhancing their day-to-day quality of life is much more emphasized.

### 4.3. Facilitating PA for Obese and Overweight Pregnant Women in Taiwan

#### 4.3.1. Increase Motivations

Our study has provided rich information about motivators (enablers) and barriers to PA among sedentary pregnant women with high BMIs. The wide range of perceived motivators and barriers are fairly consistent with identified factors in previous research in relation to women’s capability, motivation and opportunity to engage in PA [9,15,16,36,37]. Our motivators are reported regarding women’s concerns about maternal and neonatal health [37]; especially well-being of the unborn baby(s) has been frequently reported as the primary motivator for women to enhance PA awareness. “The assurance of a smooth birth” is reported as women’s top priority, similar to the highest motivator concluded from a systematic review of PA behavioral change interventions, fetal health is the greatest perceived benefit of PA during pregnancy. This personal belief increases women’s intention but does not serve alone to guarantee PA engagement, particular for pregnant women with high BMIs [2].

Another generally cited motivator for women to maintain PA during pregnancy is to manage gestational weight gain [16]. However, this weight related motivator is not as emphasized in our study as in other qualitative studies among women with high BMIs [16,37]. Another PA motivator, “*To facilitate a return to pre-pregnancy body weight and shape*” [2] is also not pervasive in our focus groups. This may be because most of our participants, recognizing the PA benefits for babies, report unlikely to engage in behavioral change soon. Their expectations on maintaining and/or returning to normal BMIs have been constantly low. To our inactive participants, since regular PA is unlikely to achieve, fetus growth and dietary choices become a higher priority than PA engagement when considering birth outcomes. The importance of healthy eating and nutrient supplements during pregnancy, not the focus of this study, is highly accentuated in the low readiness subgroup. On the other hand, participants at a higher readiness stage (*n* = 5) report gestational weight management a PA motivator. This highlights the significance of facilitating readiness to PA action; those with better readiness start to focus on their own health by perceiving and experiencing personal benefits from PA for maternal well-being.

In addition to motivators of personal beliefs, social support plays an enabling role for PA [15,20]; exploration of normative beliefs from previous PA findings suggests that women are influenced by partners, family members or friends, and by sources of information in the media and websites [2,15]. Partners’ attitudes, in contrast to other family members, have positive influences on women in PA maintenance [2]. Our participants also report leisure-time PA becomes easier if PA is supported by their partners and/or family members under the same roof. Family enablers, primarily support from husbands and parents in-law, given its cultural importance, provide relevant contexts from which to develop and implement PA interventions for Taiwanese women. We suggest PA interventions to address normative beliefs in order to create a supportive social environment within the joint family context.

#### 4.3.2. Overcome Barriers

A large number of barriers to PA participation identified in our study are broadly consistent with findings from other studies [15,20,36]. Women’s limited physical capability and opportunity to carry out PA are generally classified as barriers. When considering women’s control beliefs [2], perceived barriers may be viewed as internal (physical and psychological hindrance) and external (work, family, time and environmental constrainsHarrison and colleagues [20], after examining forty-nine articles, collecting data from 47 studies (7655 participants) have concluded that barriers, not necessarily perceived by overweight pregnant women, are predominantly intrapersonal, such as fatigue, lack of time, and pregnancy discomforts. Our most salient behavioral change barrier is also very intrapersonal that the potential harm from PA to their unborn baby is commonly brought up), including experiencing discomfort, fatigue, a lack of time, having other children priorities, and burdens from work that prevent our participants from being active [15]. In a comprehensive systematic review [7], obese pregnant women’s evidence-based barriers are in multiple domains, including pregnancy-related symptoms, lack of time, access to child care, and concerns about their safety and that of their unborn baby. Our comparable findings in Taiwan confirm that these PA barriers are universal, cross-cultural, and multi-faceted [16].

There is considerable literature indicating that pregnant women are less active than nonpregnant women and PA generally declines over pregnancy. Among the overweight and obese (N = 305), their PA declines significantly between early pregnancy and 28-week gestation, with a further decline to 36-week gestation. Likewise, women with obesity in a observational study (*n* = 155) decline their PA levels significantly in the third trimester [8]. Our participants report being less active in early and late pregnancy, contributing to fear of a possible miscarriage or premature birth, in addition to physical fatigue and discomfort [7]. In our study, being physically challenged in the early and late trimester, significant barrier to PA is “lack of motivation,” as a recognized sentiment related to low self-efficacy to change [36]. A considerable potential exists for strategized interventions to improve PA levels as high BMI women’s pregnancy advances [38]. In fact, supervised programs for overweight and obesity during pregnancy, compare with standard care, effectively help maintain PA levels and limit gestational weight gain in late pregnancy [2,38].

Other PA barriers, perceived to be outside of women’s control, are reported mostly competing priorities in terms of family and work responsibilities, such as working a full-time job, taking care of children, home maintenance, and spending time with their spouse. We observe that participants’ lack of motivation, often linked with low self-efficacy, and competing priorities occur simultaneously to negatively influence PA engagement, and such external barriers are reported as the most difficult combination of barriers.

#### 4.3.3. Strategies to Facilitate PA Readiness in Taiwan

Our findings of motivators and barriers signify the importance for obstetrician providers to discover effective and feasible strategies to address intrapersonal and social factors [20]. More research is necessary to both recognize women’s internal fear and overcoming external obstacles such as competing priorities, and studies should be designed to explore correlates regarding specific intrapersonal beliefs and external environmental barriers among inactive pregnant women. In order to support women to engage in PA in early pregnancy and postpartum, interventions are to provide women trimester-specific information on fetal protection and emphasize the potential maternal benefits from PA [2]. Practical PA advice, such as building walking regimen in daily routines and encouraging PA during the second trimester, should be provided during prenatal visits before PA gradually declines.

Previous research suggests that person-centered strategies, the incorporation of PA information and behavior change techniques, have effectively translated pregnant women’s positive attitudes into better PA participation [20]. However, one of the most important issues arising from our study is the perceived lack of accessible information and incomplete advice on the benefits and recommended levels of PA from obstetrician providers during prenatal visits. There is also concern that health professionals do not address pregnant women’s individual needs and expectations for PA [7]. Results from a mixed-method study among overweight and obese pregnant women also indicate that approximately half of women do not consider excessive gestational weight gain to be a concern during pregnancy, and this lack of knowledge has significantly affected their motivation for PA [37]. Similarly, our participants are not fully aware that their excessive weight, associated with adverse maternal health outcomes, may be reversed by PA, and knowledge regarding the infant risk is not evident during the interviews.

PA advice was described as somewhat paternalistic during the obstetrician clinic visits. While nurses and midwives are traditionally viewed as being ideally placed to advise and support women about PA in pregnancy [16], our participants in northern urban areas of Taiwan recall no advice given by nurses on PA. There exists a great difference of nurses’ advising role in between providing PA and breastfeeding advice [16]. Our participants expect to obtain useful PA information for their babies from nurses in the prenatal clinics, rather than their obstetrician doctors; in fact, sufficient time spent with clinical nurses has been brought up by two participants as necessary to effectively facilitate PA.

Health education interventions using Motivational interviewing (MI) techniques as a good behavioral counseling strategy has been successful. It helps to improve health-related self-efficacy and PA behaviors among pregnant women in a randomized controlled trial [39]. We also suggest nurse-led MI interventions be developed; employing a counseling-based approach, a MI PA intervention will be particularly beneficial for women in early pregnancy, and nurses’ mother-like image and their less paternalistic communication are to help pregnant women resolve possible ambivalence in recommended PA and neonatal outcomes and enhance women’s internal motivation for behavioral change. More research is needed to identify factors that influences pregnant women’s responsiveness to MI, such as implementation format, communication and nurse educator preferences, and clinic setting in the community. In addition, for highly educated, tech-savvy women as our participants, future interventions may focus on the potential of nurse-led e-education, combining the use of the media as a vehicle, in order to offer online or mobile-health guidance on PA throughout the entire pregnancy course. Future research will then explore the barriers and enablers among women engaging in PA facilitated by mobile APP during pregnancy and the postpartum period [9].

In terms of PA facilitation at a national level, lessons and experience from World Health Organization’s (WHO) innovative approach in Europe may shed lights to promote PA among Taiwanese women during pregnancy, given that the pandemic issue of obesity in Taiwan is comparable to what is reported in Turkmenistan. A WHO survey conducted in 2018 reveals that nearly half of the women population in Turkmenistan are overweight and 17% are obese, and the WHO European Office for the Prevention and Control of Noncommunicable Diseases and the Ministry of Health and Medical Industry of Turkmenistan have jointly developed and implemented The Turkmenistan National Strategy for Physical Activity 2018–2025 in which evidence-based PA advice is included in maternal health booklet given to all women in the country during their antenatal care visits. In order for this strategy to reach a large proportion of this group, the capacity of a national health care network is built through a ‘training of trainers’ approach to safely recommend and prescribe PA to all women during pregnancy. While interventions to encourage recommended levels of PA in pregnancy are best accompanied by accessible and consistent information about the positive maternal and birth outcomes [16], we suggest strategized PA interventions in Taiwan to incorporating policy action and call for high-level leadership by the health sector. Inspired by Turkmenistan’s comprehensive approach, further research is needed to discover culturally appropriate strategies in PA interventions to facilitate overweight/obese pregnant women’s readiness [16,38].

### 4.4. Emotional Support during Focus Groups

Our focus group seem to provide an opportunity to gather obese and overweight women based on a shared clinical diagnosis due to their high BMIs, though the purpose of this study is not to explore the nature and extent of these women’s emotional responses related to morbid obesity. Participants exhibit a unique body image reportedly formed during their puberty. A recent systematic review in adolescents showed that leisure sedentary behaviors, in our case, screen-based and stay-on the bed behaviors, are related to higher psychological distress and lower self-esteem [40]. Compared with women with healthy BMIs, our participants’ low PA [41] and negative obesity-related emotions, such as expressions and feelings of frustrations, hopeless, uselessness, and a lack of motivation to PA are clearly associated with previous unsuccessful weight management experience. Future research should examine the psychological impact of reducing time spent using screens for leisure among pregnant women, whilst accounting for possible confounding factors such as PA and popular binge-watching behaviors in Taiwan.

While evidence-based practice is suggested to prescribe PA as part of pediatric obesity treatment [41], we believe that PA advice also should be introduced and prescribed in early pregnancy with regular follow ups about PA performance. Recent systematic reviews with meta-analyses have provide evidence [41,42] that PA interventions, regardless of maternal weight [43], have a positive influence on psychological health by deducing depression, increasing self-esteem, and providing more positive body image. This promising result is particularly important to encourage PA for expectant mothers showing signs of depressions and/or negative emotions. In particular, many of our participants show self-depreciating emotions, health care providers need to pay attention and offer evidence-based PA advice, at least once a week, to reduce the symptoms and prevent depression in this period [44].

In the focus group sessions, obese and overweight participants emotionally support each other. They immediately recognize empathy and respond warm words of encouragement from other peer members. Such feelings of “*we pretend we do not care that much about our weight*: (Bella) only women with high BMIs may understand. In addition, they share deeply about pregnant experience from the expectant mothers’ perspectives; having this social opportunity to engage in PA is an enabler [15]. Being obese and pregnant, a commonality in body image, can be used by PA program developers to foster another level of sharing and mutual understanding for this group. More studies are needed to explore PA’s effects on shaping positive body image, including: what “being an obese mom” means to these women, how their body image transit or transform after pregnancy, and the way emotional support and mutual understanding from their peer have impacted high-BMI pregnant women’s PA intentions and readiness.

### 4.5. Strengths and Limitations of the Study

This qualitative design allows for an elaboration on low self-esteem and little motivated participants’ opinions and emotions for PA that would have been infeasible using survey measures. The emerging themes help to identify relevant barrier management and behavioral support strategies that could be included in the PA intervention design. The study emphasis on perceptions and readiness for PA, employing a TTM theoretical approach, are first-hand information from high BMI, sedentary pregnant women’s perspectives in Taiwan. Insights are provided by this study using various steps in the analysis process to ensure reliability of this study; useful, preferrable strategies are explored for developing effective PA intervention programs. In addition, our participants are relatively educated and tech-savvy, their potential and familiarity in using wearable device and smartphones provides further opportunities for program developers to design a more efficient and personal PA intervention using strategies of mobile-Health APP that may be beneficial for residents not only in urban but also remote areas in Taiwan.

A few limitations must be acknowledged while interpreting our findings of a focus group study, but the main one is the use of a small convenient sample of highly educated, unemployed, tech-savvy pregnant women with high BMIs living in metropolitan and urban areas which constrains the generalizability of our findings. In addition, although those who have participated in PA programs are excluded, our participants may have already been motivated to manage weight and maintain a healthy lifestyle before pregnancy. The motivations and barriers to PA engagement may be inevitably related to their previous weight management experience. Other biases derived from the relatively inflexible semi-instructed interview guide, the moderators’ experience, the dynamics of the interviews, and the coders’ subjective analyses [16] also rise to affect the reliability and validity of this study. Despite these limitations, it is essential to conduct such qualitative studies sampling various obese and overweight groups of pregnant women. This allows descriptions and analyses of detailed rationales behind behavioral decision-making about PA engagement, “*aiding the accuracy and communication required to build a cumulative evidence base”* ([2], p. 18).

## 5. Conclusions

Low levels of PA are a health concern among women with a sedentary lifestyle, and being overweight or obese during pregnancy is associated with increased risks of maternal and fetal health complications. While PA benefits in pregnancy are well-recognized with potential opportunities to implement PA interventions and effectively improve birth-related outcomes, little is known about overweight and obese women’s perceptions of PA and how they may be further facilitated for an active lifestyle. Success requires greater understanding of these women’s beliefs, intention, and particularly their motivations and barriers to engage in PA. Our focus group study employs a series of in-depth qualitative interviews, guided by a behavioral change model of the Transtheoretical Model, to apprehend overweight and obese women’s rationales behind PA engagement during pregnancy and postpartum.

A detailed description of motivators of and barriers to PA engagement is provided by a group of highly educated, technology-savvy women residing in urban areas in Taiwan. On a theoretical basis, a total of fourteen themes emerged and structured at five higher concept levels. Our findings illustrate the extent to which obese and overweight pregnant women’s generally low readiness for PA is commonly affected by multiple factors, including personal beliefs, perceived societal norms, peer support, and the uncontrolled competing priorities in the environment. Further research is necessary to identify effective strategies to anticipatedly prevent PA declines as pregnancy advances. In addition, obstetrician health professionals should address high BMI women’s individual needs and employ a fetus-centered approach to provide accessible information necessary for increasing motivations, overcoming barriers, and providing peer support to facilitate overweight and obese women’s readiness for PA. Culturally appropriate strategies are key to develop group-based interventions and promote a healthy lifestyle postpartum.

## Figures and Tables

**Figure 1 ijerph-18-05275-f001:**
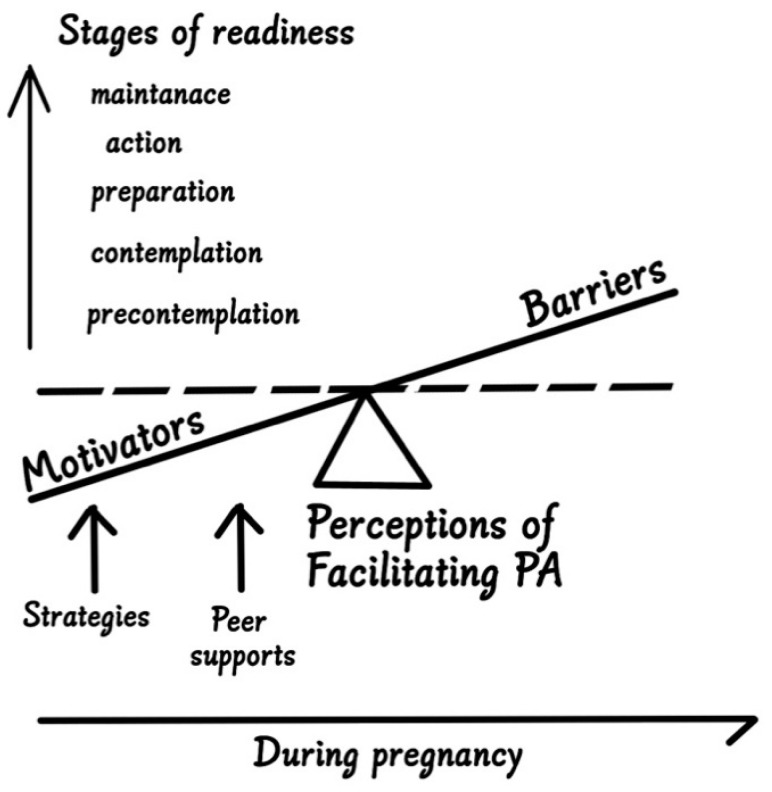
The concept of facilitating readiness for physical activity among overweight and obese women during pregnancy.

**Table 1 ijerph-18-05275-t001:** Questions in the Semi-structured Interview Guide.

1. How do you manage your weight gain/loss before and during your pregnancy?
2. Have you had successful experience to manage your weight by increasing physical activity?
3. When you hear people say, “increase your physical activity” or “be more active in your leisure time,” what comes to your mind? What is your impression when people talk about the connection between weight and physical activity?
4. What have your obstetrician suggested regarding physical activity, if any?
5. How much do you intend to change your current lifestyle? Are you willing to start increasing physical activity? Why and why not?
6. Do you think it is necessary now to adopt a more active lifestyle? Say, increase PA in your leisure time?
7. Do you want to know more about how to increase physical activity?
8. Are you engaged in any physical activity routine? Official or unofficial? How?
9. Say if you plan to be more active on a regular basis, what would be the possible hurdle?
10. If you have increased your physical activity during pregnancy, do you plan to continue doing so?
11. Do you need to get someone’s approval or assistance in order to achieve your physical activity plan? Who would that be?
12. What would be considered appropriate advice for an obstetrician or a nurse to tell his/her patient to increase physical activity, what is your suggestion?
13. How do you prefer to be motivated or assisted to increase physical activity after you deliver your baby?

**Table 2 ijerph-18-05275-t002:** Characteristics of Pregnant Participants.

	*n*	%
Pre-pregnancy BMI		
Overweight (≥25)	7	53.9
Obese (≥30)	6	46.1
Parity		
nulliparous	9	69.2
multiparous	4	30.8
Employment		
Yes	4	30.8
No	9	69.2
Married		
Yes	11	84.6
No	2	15.4
Education level		
College or university	13	100
Smartphone possession		
Yes	13	100
Online search for pregnant information		
Yes	13	100
Experience of using pregnant-related APPs		
Yes	7	53.9
No	6	46.1
Being suggested to manage gestational weight gain		
Yes	13	100
Being suggested to increase physical activity		
Yes	13	100

**Table 3 ijerph-18-05275-t003:** Emerging Focus Group Themes.

I. Perceptions of Physical Activity Engagement	
Theme 1	Physical activity perceived as a cliché when being suggested by obstetricians
Theme 2	Physical activity triggers negative experience about unsuccessful weight management
Theme 3	Engaging in higher levels of physical activity evokes negative emotions (representing an impossible task)
II. Motivation and Readiness to Adopt an Active Lifestyle	
Theme 4	All obese and overweight pregnant women are believers of PA’s beneficial effects
	Lower levels of motivation to change are associated with lower levels of self-esteem and self-efficacy
	Higher levels of motivation to change are associated with fetus health, child-bearing responsibilities, and significant others’ expectations
Theme 5	A tongue never gets things done and preparation usually takes a long time
	Information is needed to prepare for pregnancy-appropriate physical activities
Theme 6	Higher levels of readiness are associated with successful weight management experience
III. Barriers to Engage in Higher Levels of Physical Activity	
Theme 7	Particular cultural beliefs limit pregnant women’s physical activity
Theme 8	Fear of birth-related complications limits pregnant women’s physical activity
Theme 9	Low peer support (to work out together) and limited support from significant others
	External conditions perceived inappropriate for physical activities
	Competing family and/or work priorities
IV. Preference of Facilitation Strategies	
Theme 10	Empathetic communication from the health care providers
Theme 11	A “fetus-centered” approach to motivate mothers and facilitate physical activity
	“Patient-centered” counseling to overcome barriers by tailoring each individual’s needs
V. Peer Support During Focus Groups	
Theme 12	Support for being an expectant mother
Theme 13	Support for low self-esteem: empathizing similar body image and unsuccessful weight management (being lazy and ugly)
Theme 14	Support for low self-efficacy: empathizing similar feelings of frustration regarding multiple barriers to initiate an active lifestyle

## Data Availability

The data presented in this study are available on request from the corresponding author. The data are not publicly available due to funding restrictions and patient privacy.

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
