# Peer review of "“Moving for My Baby!” Motivators and Perceived Barriers to Facilitate Readiness for Physical Activity during Pregnancy among Obese and Overweight Women of Urban Areas in Northern Taiwan"

_ijerph, 2021, doi:10.3390/ijerph18105275_

Round 1
Reviewer 1 Report
The qualitative article focuses on the study of a sample of pregnant women in northern Taiwan, at risk of personal health and their fetus, due to their obesity, sedentary lifestyle and low levels of physical activity (PA).
In this regard, I make some observations and observations:
Summary and keywords:
The abstract includes the necessary aspects that determine the general vision of the manuscript, its background, the purpose (objective) of the study, the methods, results and an objective representation that does not exceed the main conclusions.
- Introduction:
With a broad and understandable context, the importance of the study is highlighted, the purpose of the work and its meaning are defined, reviews are made on the current state of the research field, key and important publications are cited. Similarly, reference is made to the hypothesis that will be tested in the study. The main objective of the work is clear, and the main conclusions are highlighted.
- Materials and methods:
In this section, the obtaining of the informed consents is confirmed. The method, data and associated protocols are described in sufficient detail. The study also indicates the exclusion of women with clinical diagnoses of depression or psychiatric disorders. Sufficient description facilitates subsequent replication from the published results.
- Results
The table with the characteristics of the participants facilitates the understanding of the results, as well as the table with the 14 emerging themes that focus on the perceptions, motivation, difficulties and preferences of the women participating in the study. The figure also facilitates and clarifies the concept of preparation for physical activity among overweight and obese women during pregnancy.
The characteristics of the participants, their perceptions of motivation and disposition for physical activity, as well as the procedures, instruments, and data are satisfactorily indicated for their analysis with reference to reliability.
Strategies that have to do with the new lifestyle, communication and support are highlighted, with reference to the high level of motivation of PA for the health of their future babies, to improve delivery and maternal health.
Similarly, barriers and difficulties for PA are evident, specifying in detail some at a personal and social level, such as low support from colleagues and other significant others, as well as at an environmental level, highlighting heat, fatigue and daily work. among others.
The manuscript adequately details the need for advice, motivation and support to promote PA in the sample group. It is necessary to highlight the two-way communication style, the personalized attention and assistance of the experts, the elimination of personal and environmental barriers, to design a customized physical activity program.
It is evidenced in the qualitative study how pregnant women show empathy, share their emotions, joys, worries, frustrations and negative experiences.
- Discussion
Results, findings and implications are discussed in a broad context and interpreted in perspective with previous studies: preparation for action, counseling on PA, etc.
It is satisfactorily reported on the increase in motivations, seeking the well-being and benefit of the health of the fetus, the control of gestational weight gain and the safety of a smooth delivery. It is reported that the greater the preparation for the control of gestational weight, the greater the motivation for PA. This highlights the importance of facilitating preparation for the activity.
It is indicated that social support is another great facilitator for action with FA, in the influence of their partners, family, friends and other sources and media. In free time, this social support and other relevant contexts are perceived more.
Barriers to physical activity for pregnant mothers are discussed, keeping quite in line with those of other studies. They refer to internal barriers (physical and psychological impediment) and external barriers (work, family, time and environmental situations). These barriers are revealed to be universal, cross-cultural, and multifaceted. To achieve the promotion of physical activity in this group, it is necessary to apply the WHO strategy by training all women during pregnancy.
The strengths of the study are highlighted: qualitative design, emerging issues and its theoretical approach; strategies for developing effective intervention programs; the technological training of the participants, etc.
The limitations of the work are listed, as well as the possibility of continuing the research with qualitative studies that help with the precision and communication necessary to build an accumulated evidence base.
- Conclusions
It is concluded by pointing out that low levels of PA are a health problem among women with a sedentary lifestyle and overweight during pregnancy, associated with an increased risk of complications in maternal and fetal health, as a hypothesis test. Little is known about how active lifestyles can be enhanced by these women. Success requires a greater understanding of beliefs, motivations, and barriers to participating in physical activity.
The references seem adequate to me, although there are some self-citations by the authors.
I congratulate the authors for their detailed and clear work, appropriate to the style of the scientific article, providing the knowledge and reports that show the health benefit, through physical activity, of obese pregnant women and their future babies. This will facilitate the development of specific programs for this purpose.
Author Response
Thank you for your valuable time. On behalf of the authors, I have provided point-by-point explanations below to respond to your comments.
Point 1: The reviewer kindly provided his/her observations, and sections of introduction, materials and methods, results, discussion and conclusion of our manuscript seem to meet the standard and in general, satisfactorily answer to the research questions.
By quoting the reviewer’s final comments, we determine that no changes are necessary. “I congratulate the authors for their detailed and clear work, appropriate to the style of the scientific article, providing the knowledge and reports that show the health benefit, through physical activity, of obese pregnant women and their future babies. This will facilitate the development of specific programs for this purpose.”
Reviewer 2 Report
Moving for My Baby!” Motivators and Perceived Barriers to 2 Facilitate Readiness for Physical Activity During Pregnancy 3 Among Obese and Overweight Women of Urban Areas in 4 Northern Taiwan
The authors explored the motivators and enablers, as well as the perceived barriers to physical activity (PA) participation among Taiwan obese and overweight pregnant women. The study specifically provide qualitative insights into motivators, environmental enablers, and the perceived barriers of PA. Given the low levels of prenatal physical activity across countries, such information is important to guide context-specific interventions during pregnancy and the postpartum phase.
The manuscript is scientifically, well-written and informative.
Only few queries noted, particularly in the abstract.
Abstract
Lines 13-14: “Low levels of physical activity (PA) are of a health concern among high body mass indexes (BMIs) women with a sedentary lifestyle, and being overweight or obese during pregnancy is associated with increased risks of maternal and fetal health complications.” I suggest rephrasing this sentence to read: ….high body mass index of women living a sedentary lifestyle, and are overweight or obese during pregnancy….
Line 19: Delete the word ‘greater; in the sentence. Also, the sentence should read…pregnant women’s decision to engage, or refrain from PA practice.
Line 19: Delete the ‘the total of’. Start the sentence with ‘Thirteen…..
Lines 22-23: The sentence is complex. Break it down. The phrase ‘to inductively categorize’ should be rephrase for clarity.
Line 24: Start the sentence with ‘The findings….Delete “Our qualitative findings”. The study by design is a qualitative study, so there is no need for ….
Lines 26-27: The phrasing “are to be effective focusing” is not grammatically correct.
Line 29: …. “to change”. This is incomplete sentence. What kind of change. Qualify the change.
Introduction
Line 58: “The 2015 guide- 58 lines published by the American College of Obstetrics and Gynecology (ACOG)… I suggest you cite the latest version, which is 2020 ACOG guideline.
Author Response
Thank you for your valuable time. On behalf of the authors, I have provided point-by-point explanations below to respond to your comments.
Point 1: In general, the reviewer gave positive feedbacks, “The manuscript is scientifically, well-written and informative.”
Point 2: We have addressed queries pointed out by reviewer 2, mainly in the abstract.
Reviewer’s comments |
Changes (in red) in the manuscript revision |
1. Lines 13-14: I suggest rephrasing this sentence to read: …. high body mass index of women living a sedentary lifestyle, and being overweight or obese during pregnancy…. |
Line 13-14: Low levels of physical activity (PA) are of a health concern among high body mass indexes (BMIs) of women living a sedentary lifestyle, and being overweight or obese during pregnancy is associated with increased risks of maternal and fetal health complications. |
2. Line 19: Delete the word ‘greater; in the sentence. Also, the sentence should read…pregnant women’s decision to engage, or refrain from PA practice. |
Line 19: The purpose of this study is to explore motivators/enablers and perceived barriers through in-depth qualitative inquiry, guided by a behavioural change theory, for greater understanding of pregnant women’s rationales behind PA engagement decisions to engage, or refrain from PA practice. |
3. Line 19: Delete the ‘the total of’. Start the sentence with ‘Thirteen….. |
Line 22: A total of Thirteen overweight and obese pregnant women… |
4. Lines 22-23: The sentence is complex. Break it down. The phrase ‘to inductively categorize’ should be rephrase for clarity. |
Line 24: A comprehensive thematic content analysis was performed with a constant comparison method to inductively categorize interview data and generate themes. |
5. Line 24: Start the sentence with ‘The findings….Delete “Our qualitative findings”. The study by design is a qualitative study, so there is no need for …. |
Line 26: Our qualitative The findings illustrate the extent… |
Reviewer’s comments |
Changes (in red) in the manuscript revision |
6. Lines 26-27: The phrasing “are to be effective focusing” is not grammatically correct. |
Line 28: PA interventions are to be effective by focusing on… |
7. Line 29: …. “to change”. This is incomplete sentence. What kind of change. Qualify the change. |
Line 29: Strategies shared by participants shed lights for program developers to design preferable behavioral interventions for this group of women who are low self-esteem with low self-efficacy to increase PA and meet recommended levels. |
8. Line 58: “The 2015 guideline published by the American College of Obstetrics and Gynecology (ACOG)… I suggest you cite the latest version, which is 2020 ACOG guideline. |
Line 64: The reference [14] has been changed to cite the most updated 2020 ACOG guideline.
14. Physical activity and exercise during pregnancy and the postpartum period. American College of Obstetricians and Gynecologists (ACOG) Committee Opinion No. 804. Obstet Gynecol 2020, 135, e178–88. |